# The Structure of the “Vibration Hole” around an Isotopic Substitution—Implications for the Calculation of Nuclear Magnetic Resonance (NMR) Isotopic Shifts

**DOI:** 10.3390/molecules25122915

**Published:** 2020-06-24

**Authors:** Jürgen Gräfenstein

**Affiliations:** Department of Chemistry and Molecular Biology, University of Gothenburg, SE-412 96 Gothenburg, Sweden; jurgen.grafenstein@chem.gu.se

**Keywords:** NMR isotopic shifts, difference-dedicated vibrational perturbation theory, cyclic and polycyclic hydrocarbons, halonium-bonded complexes, intra-molecular hydrogen bonds, salicyl aldehyde derivatives

## Abstract

Calculations of nuclear magnetic resonance (NMR) isotopic shifts often rest on the unverified assumption that the “vibration hole”, that is, the change of the vibration motif upon an isotopic substitution, is strongly localized around the substitution site. Using our recently developed difference-dedicated (DD) second-order vibrational perturbation theory (VPT2) method, we test this assumption for a variety of molecules. The vibration hole turns out to be well localized in many cases but not in the interesting case where the H/D substitution site is involved in an intra-molecular hydrogen bond. For a series of salicylaldehyde derivatives recently studied by Hansen and co-workers (*Molecules*
**2019**, *24*, 4533), the vibrational hole was found to stretch over the whole hydrogen-bond moiety, including the bonds to the neighbouring C atoms, and to be sensitive to substituent effects. We discuss consequences of this finding for the accurate calculation of NMR isotopic shifts and point out directions for the further improvement of our DD-VPT2 method.

## 1. Introduction

Many electronic quantities of a molecule undergo small but specific changes upon isotopic substitution [1]. Such isotope effects have been observed and investigated for global properties such as dipole moments [2,3] and polarizabilities [4,5] but are most interesting for atom-specific spectroscopic properties such as electron spin resonance (ESR) [6,7], spin rotation constants [8,9] and above all nuclear magnetic resonance (NMR) isotopic shifts, in particular on chemical shielding constants [10,11,12,13,14,15]. NMR isotopic shifts have been used to investigate inter- and intramolecular hydrogen bonds [14,15,16,17,18,19,20] and salt bridges [21,22] as well as the backbone conformation in proteins [23,24,25] and local stereochemistry [26,27,28,29].

Isotopic effects on electronic properties are connected to the molecular vibrations—the observed value of the property is a vibrational average over the property surface around the electronic equilibrium geometry. An isotopic substitution changes the vibration motif, and this change is reflected in a change of the observed value. For a single isotopic substitution, the change of the vibration motif is expected to be localized in the region around the substitution site, and for the most common case of a substitution with a heavier isotope, the vibrational amplitude in this region will decrease. This situation resembles the relationship between the electronic energy and the exchange and correlation (XC) hole in electronic-structure theory; therefore, we have coined the term “vibration hole” [30] for the region where the vibration motif is changed substantially by the isotopic substitution.

In molecules that rapidly interconvert between different tautomers, the isotope effect described in the previous paragraph (commonly called intrinsic) is complemented by the so-called equilibrium contribution, which is due to the isotope effect on the zero-point vibration energies and in extension the Boltzmann equilibrium between the tautomers [31]. Equilibrium isotopic shifts have been used by Perrin and co-workers [32,33,34] to study the properties of intramolecular hydrogen bonds and by Ohta [35] and Erdélyi and co-workers [36,37,38,39,40,41,42] to study the molecular potential around three-center four-electron halonium bonds. However, the scope of the present work is restricted on the intrinsic contributions.

An efficient and reliable method to theoretically predict NMR isotopic shifts will facilitate their usefulness in chemical investigations. Such a method needs to account for the quantum nature of the nucleus system. Multi-component (MC) density-functional theory (DFT) [43] uses a full quantum-mechanical description of all or selected nuclei and has been used for example, to determine NMR isotopic shifts in aromatic hydroxy acyl compounds [44] Recently, NMR isotopic shifts were calculated incorporating nuclear quantum effects by path-integral molecular dynamics (PIMD) [45,46,47]. However, most calculations of NMR isotopic shifts start from the Born-Oppenheimer approximation [48] and scan potential-energy and property surfaces of the molecule around its equilibrium geometry. A widely used approach for this purpose is second-order vibrational perturbation theory (VPT2) starting from the harmonic normal vibrations of the molecule, which was presented by Kern [49] and put into an applicable form, among others, by Åstrand, Ruud and co-workers [50,51]. Jameson [52] worked out the expressions for NMR isotopic shifts in this framework, and a number of authors have performed calculations of NMR isotopic shifts at this level of theory [53,54,55] or more elaborate versions of the algorithm [56].

For a molecule with *N* atoms, the vibrational averaging in such a calculation requires to compute the chemical shielding constants of the molecule for 2(3N−6) slightly distorted geometries (2 calculations per normal mode), which totally amounts to 12N−23 calculations (including that for the equilibrium geometry). For increasing *N*, the cost of such a calculation increases rapidly, and calculations of this kind have rarely been performed for other than relatively small molecules. To overcome this limitation, a number of methods have been developed based on the *a priori* assumption that the vibration hole is localized around the substitution site [57] and the vibrational averaging can be reduced to this part of the molecule. We will call these methods *a priori* local methods in the following. For the common case of a H/D substitution, in particular, it was assumed that the vibration hole only affects the movement of the H/D atom and thus the geometry parameters for the bond involving this atom. The simplest type of *a priori* local method considers only the bond-length contraction for the bond containing the H/D atom [21,57,58], other approaches take also the decrease of the vibration amplitudes into account and involve both stretching and bending vibrations [59]. One of the most elaborate and systematic methods developed in this spirit is the Local Mode Zero-Point Level (LMZL) Approach by Yang and Hudson [60], which gives results in fairly good agreement to experiment. *A priori* local methods require the calculation of the chemical shielding constants only for a limited number of geometries (typically 10 or less) and make calculations of NMR isotopic shifts feasible for relatively large molecules. However, to the best of our knowledge, the structure of the vibration holes has never been investigated systematically, and it is not verified whether, or for which kind of compounds, it is justified to assume the vibration hole to be localized.

In a recent paper [30], we presented the difference-dedicated (DD) VPT2 approach, which provides values for NMR isotopic shifts (or isotope effects on other quantities) at VPT2 quality, that is, without any assumptions on the structure of the vibration hole, at a cost comparable to the LMZL or other comparable *a priori* local methods. As a by-product, which was not addressed in Reference [30], the DD-VPT2 approach efficiently provides information on the structure of the vibration hole. In the present paper, we will make use of this possibility and investigate the structure and locality of the vibration hole both for the systems investigated in Reference [30] and for a number of molecules studied recently by Hansen et al. [20] where the substitution site is involved in an intra-molecular hydrogen bond. The systems under consideration will be studied both with DD-VPT2 and a local VPT2 version (loc-VPT2), which is used as a model *a priori* local method (see Section 2.2 for details). For selected systems, the DD-VPT2 and loc-VPT2 results are additionally compared to LMZL results.

The paper is organized as follows: in Section 2, the DD-VPT2 and loc-VPT2 methods are described in brief, and the geometry parameters used to describe the vibration hole are defined. Section 3 presents the results of our study—in Section 3.1, we will reanalyze and complement the results for the cyclic and polycyclic molecules studied in Reference [30] and in Section 3.2, those for the halonium complexes investigated in Reference [30]. Section 3.3 is dedicated to the investigation of the salicyl aldehyde derivatives originally investigated in Reference [20]. Section 4 describes the computational methods in more detail. Section 5 summarizes the conclusions of our work and presents an outlook on future directions of investigation. Some technical details are treated in two appendices.

## 2. Theory

We define the isotope effect on NMR chemical shifts as
(1a)$$\begin{array}{ccc} {}^{n}\Delta \delta & = & \delta^{\left(B\right)}-\delta^{\left(A\right)} \end{array}$$
(1b)$$\begin{array}{cc} = & \sigma^{\left(A\right)}-\sigma^{\left(B\right)}, \end{array}$$
where *n* is the number of bonds between the substitution site and A and B are the unsubstituted and the substituted isotopologues, respectively. Where appropriate, the NMR isotope shift at reporting nucleus Q will be denoted by ${}^{n}\Delta \left(Q\right)$. We will also use the notation ${}^{n}\Delta$ (with unspecified *n*) for isotope effects on other properties, for example, ${}^{n}\Delta r$ for the isotope effect on bond length r. In contrast, Δ without a superscript to the left will denote vibration amplitudes and vibration corrections.

### 2.1. Standard Versus Difference-Dedicated Second-Order Perturbation Theory

In standard VPT2, the NMR isotope shift between isotopologues A and B is represented as the sum of a harmonic part connecting the curvature of the chemical-shielding surface with the harmonic vibrations, and an anharmonic part connecting the anharmonic part of the vibrations (i.e., the change of the average geometry) with the gradient of the σ surface: (2)$$\begin{array}{ccc} {}^{n}\Delta \delta & = & \Delta \delta^{\left(B\right)}-\Delta \delta^{\left(A\right)}, \end{array}$$(3)$$\begin{array}{ccc} \Delta \delta^{\left(X\right)} & = & \Delta \delta_{\mathrm{harm}}^{\left(X\right)}+\Delta \delta_{\mathrm{anh}}^{\left(X\right)}, \end{array}$$(4)$$\begin{array}{ccc} \Delta \delta_{\mathrm{harm}}^{\left(X\right)} & = & -\frac{1}{4}\sum_{i}^{v}D^{2}\sigma \left({\underline{l}}_{i}^{\left(X\right)}\right), \end{array}$$(5)$$\begin{array}{ccc} \Delta \delta_{\mathrm{anh}}^{\left(X\right)} & = & -D\sigma \left(\Delta {\underline{R}}^{\left(X\right)}\right), \end{array}$$(6)$$\begin{array}{ccc} \Delta {\underline{R}}^{\left(X\right)} & = & -\frac{1}{4}\sum_{i}^{v}\frac{1}{\omega_{i}}\sum_{k}^{v}D^{1,2}V\left({\underline{l}}_{i}^{\left(X\right)},{\underline{l}}_{k}^{\left(X\right)}\right){\underline{l}}_{i}^{\left(X\right)}. \end{array}$$

Here, $\omega_{i}^{\left(X\right)}$ and ${\underline{l}}_{i}^{\left(X\right)}$ are the harmonic frequencies and Cartesian displacement vectors for the normal vibration modes of isotopologue *X* (X=A,B), where $i=1,2,\ldots n_{\mathrm{vib}}$, $n_{\mathrm{vib}}=3N-6$ (3N−5 for linear molecules, *N* is the number of atoms), and the index v stands for vibrational modes. Here and in the following, bold print marks Cartesian vectors (slanted) or tensors (not slanted), single or double underlining, vectors or tensors over the set of nuclei. We will use dimensionless vibrational coordinates [61] and correspondingly scaled vibration modes throughout this paper. Furthermore, we employ directional derivatives defined according to
(7a)$$\begin{array}{ccc} D^{n}C\left(\underline{l}\right) & = & {\left[\frac{\partial^{n}}{\partial \lambda^{n}}C\left(\lambda \underline{l}\right)\right]}_{\lambda =0}, \end{array}$$
(7b)$$\begin{array}{ccc} D^{n_{1},n_{2}}C\left({\underline{l}}_{1},{\underline{l}}_{2}\right) & = & {\left[\frac{\partial^{n_{1}+n_{2}}}{\partial \lambda_{1}^{n_{1}}\partial \lambda_{2}^{n_{2}}}C\left(\lambda_{1}{\underline{l}}_{1}+\lambda_{2}{\underline{l}}_{2}\right)\right]}_{\lambda_{1}=\lambda_{2}=0}, \end{array}$$ (and $D\ldots =D^{1}\ldots ).$ In practical calculations, the directional derivative in Equation (5) is usually decomposed into components from the ${\underline{l}}_{i}^{\left(X\right)}$. The vector $\Delta {\underline{R}}^{\left(X\right)}$ describes the change of the $r_{z}$ geometry [62] for isotopologue *X* due to the anharmonic vibrations. That is, the nuclear coordinates for the $r_{z}$ geometry are given by ${\underline{R}}_{z}^{\left(X\right)}={\underline{R}}_{e}+\Delta {\underline{R}}^{\left(X\right)}$ where the vector ${\underline{R}}_{e}$ collects the nuclear coordinates for the $r_{e}$ geometry.

To evaluate Equations (Equation 2) *et seq.*, for each isotopologue one has to calculate $n_{\mathrm{vib}}$ second-order directional derivatives of σ (Equation (4)) and the complete semi-diagonal cubic force constant matrix (Equation (6)), which in turn requires the calculation of the chemical shielding constants and the harmonic force constant matrix for $2n_{\mathrm{vib}}$ slightly distorted geometries each per isotopologue. This numerical cost increases rapidly with increasing size of the molecule.

The DD-VPT2 method presented recently by the author of Reference [30] allows to calculate NMR chemical shifts at VPT2 quality but with a limited number (typically less than 20) calculations of σ and the force-constant matrix. At the heart of the system is a set of non-canonic vibration modes common to the pair of isotopologues, called difference-dedicated vibration modes, that are constructed such that they reflect the structure of the vibration hole as efficiently as possible. These vibration modes are calculated in the following way:1Determine the $\omega_{i}^{\left(X\right)}$ and the ${\underline{l}}_{i}^{\left(X\right)}$ for A and B. Complement each set of ${\underline{l}}_{i}^{\left(X\right)}$ by three modes for rigid translations along the principal axes of inertia of (X) and 3 (or 2) modes for rigid rotations around the principle axes of inertia. The scaling of these additional modes is arbitrary.2Find the coefficient matrix $J_{ij}$ for the expansion
(8)$${\underline{l}}_{i}^{\left(B\right)}=\sum_{i^{\prime }}^{v+r+t}J_{ii^{\prime }}{\underline{l}}_{i^{\prime }}^{\left(A\right)},$$(the *J* stands for Jacobian). Here, the indices v, r, and t stand for vibrational, rotational, and translational modes, respectively.3Construct the matrix $\left(\left(M_{kk^{\prime }}\right)\right)$ according to
(9)$$M_{kk^{\prime }}=\sum_{i}^{v+r}J_{ik}J_{ik^{\prime }}-\Theta_{v}\left(k\right)\delta_{kk^{\prime }},$$
where $\Theta_{v}\left(k\right)$ is 1 if *k* refers to a vibration mode and 0 otherwise.4Diagonalize the matrix $\left(\left(M_{kk^{\prime }}\right)\right)$, denote the eigenvalues as $\kappa_{i}$, which are ordered such that the $\left|\kappa_{i}\right|$ decrease monotonously. Collect the eigenvectors as column vectors in the matrix $\left(\left(K_{ij}\right)\right).$5Calculate the difference-dedicated vibration modes ${\underline{l}}_{i}^{\left(\Delta \right)}$ according to
(10)$${\underline{l}}_{i}^{\left(\Delta \right)}=\sum_{k}^{v+r}K_{ik}{\underline{l}}_{k}^{\left(A\right)}.$$

The weight factor $\kappa_{i}$ indicates how much the vibration along ${\underline{l}}_{i}^{\left(\Delta \right)}$ differs in its amplitude between isotopologues A and B, that is, how much vibration mode *i* contributes to the vibration hole. It has turned out [30] that for a single isotopic substitution, the first three $\kappa_{i}$ are roughly $-1+\sqrt{m(A)/m(B)}$ where m(A,B) are the masses of the two isotopes, that is, about −0.2926 for a H/D substitution and −0.0393 for a ${}^{12}$C/${}^{13}$C substitution. The corresponding three vibration modes are localized around the substitution site. For i>3, the $\left|\kappa_{i}\right|$ decay rapidly; typically fewer that the first ten of them are >$10^{-3}$. For multiple isotopic substitutions, one typically finds the corresponding numbers of relevant difference-dedicated vibration modes per substitution. That is, the cost for a DD-VPT2 calculation is roughly proportional to the number of substitution sites, and the gain in computation time compared to a standard VPT2 calculation will be less pronounced (but in most cases still substantial) for systems with multiple isotopic substitutions.

With the difference-dedicated vibration modes, the NMR isotopic shifts can then be calculated as follows: (11)$$\begin{array}{ccc} {}^{n}\Delta \delta & = & {}^{n}\Delta \delta_{\mathrm{harm}}+{}^{n}\Delta \delta_{\mathrm{anh}}, \end{array}$$(12)$$\begin{array}{ccc} {}^{n}\Delta \delta_{\mathrm{harm}} & = & -\frac{1}{4}\sum_{k}^{v+r}\kappa_{k}D^{2}\sigma \left({\underline{l}}_{k}^{\left(X\right)}\right), \end{array}$$(13)$$\begin{array}{ccc} {}^{n}\Delta \delta_{\mathrm{anh}} & = & -D\sigma \left({}^{n}\Delta \underline{R}\right), \end{array}$$(14)$$\begin{array}{ccc} {}^{n}\Delta \underline{R} & = & -\frac{1}{4}\sum_{k}^{v+r}\kappa_{k}\sum_{ii^{\prime }}^{v+r}G_{ii^{\prime }}D^{1,2}V\left({\underline{l}}_{i}^{\left(\Delta \right)},{\underline{l}}_{k}^{\left(\Delta \right)}\right){\underline{l}}_{i^{\prime }}^{\left(\Delta \right)}, \end{array}$$(15)$$\begin{array}{ccc} G_{ii^{\prime }} & = & \sum_{j}^{v}K_{ji}K_{ji^{\prime }}\frac{1}{\omega_{j}^{\left(A\right)}}. \end{array}$$

Owing to the rapid decay of the $\left|\kappa_{i}\right|$, only the first few terms in Equation (Equation 12) and only the first few directional derivatives with respect to ${\underline{l}}_{k}^{\left(\Delta \right)}$ need to be calculated; besides, the calculation need not be done twice (for each of the isotopologues). As a consequence, DD-VPT2 may save 90 % or even more computation time as compared to standard VPT2 [30].

The ${}^{n}\Delta {\underline{R}}^{\left(\Delta \right)}$ describe the isotope effect on the $r_{z}$ geometry. One can construct a fictitious geometry ${\underline{R}}_{e}+{}^{n}\Delta {\underline{R}}^{\left(\Delta \right)}$, which has no immediate physical meaning but when compared to ${\underline{R}}_{e}$ reflects the isotope effect on the $r_{z}$ geometry. We will also use geometries of the form ${\underline{R}}_{e}+\lambda {\;}^{n}\Delta {\underline{R}}^{\left(\Delta \right)}$ with λ being a dimensionless number ≫1, which in comparison to ${\underline{R}}_{e}$ allow to visualize the isotope effect on the geometry.

### 2.2. Local Second-Order Vibrational Perturbation Theory (Loc-Vpt2)

For test purposes, we defined an *a priori* local version of standard VPT2 (loc-VPT2). In loc-VPT2, only the nucleus at the substitution site is treated as in standard VPT2 while all other nuclei are kept fixed at their $r_{e}$ positions. Except from this modification, the loc-VPT2 values for ${}^{n}\Delta \delta$ will be calculated in the same way as in Equation (Equation 2). We note that in loc-VPT2, the local vibration modes will be found by diagonalizing the restricted force-constant matrix, that is, in distinction to many other *a priori* local methods, there are no explicit assumptions on the vibration motif, and harmonic couplings between for example, stretching and bending modes will be taken into account properly. A local version of the vibration hole can be obtained from loc-VPT2 in the same way as for standard VPT2.

### 2.3. Geometry Parameters

While a full description of the vibration hole requires to compare the vibrational wave functions for the two isotopologues its most relevant features can be described effectively by changes in the geometry and vibrational amplitudes.

Unlike the equilibrium geometry ${\underline{R}}_{e}$, the effective geometry of a vibrating molecule is not defined unambiguously. A common definition is the $r_{z}$ geometry [62], which is given by
(16)$${\underline{R}}_{z}^{\left(X\right)}={\langle \underline{R}\rangle }_{\left(X\right)},$$
where ${\langle \ldots \rangle }_{\left(X\right)}$ indicates averaging over the vibrational wave function for isotopologue *X*. In distinction, the $r_{g}$ geometries [62] are not derived from a set of effective Cartesian coordinates; instead, each geometry parameter is found by an individual vibrational averaging. If the geometry parameter *f* is expressed as a function $f\left(\underline{R}\right)$ of the nuclear coordinates then
(17)$$f_{g}={\langle f\left(\underline{R}\right)\rangle }_{\left(X\right)}.$$

For comparison,
(18)$$\begin{array}{ccc} f_{e} & = & f\left({\underline{R}}_{e}\right),\\ f_{z}^{\left(X\right)} & = & f\left({\underline{R}}_{z}^{\left(X\right)}\right) \end{array}$$
(19)$$\begin{array}{cc} = & f\left({\langle \underline{R}\rangle }_{\left(X\right)}\right). \end{array}$$

While the $r_{g}$ geometry parameters give the most natural description of geometry changes and are most closely related to the NMR mechanism the $r_{z}$ values are connected to the calculation of NMR isotopic shifts in the VPT2 or DD-VPT2 framework and sometimes prove most appropriate for the description of long-range geometry effects (see e.g., Section 3.3).

The mean-square amplitude for geometry parameter *f* in harmonic approximation is given by $\langle \Delta f^{2}\rangle$, taken with the respective vibrational wave function. This value is given with reference to the $r_{e}$ geometry. In addition, the covariance $\langle \Delta f\Delta g\rangle$ for different geometry parameters *f* and *g* is in general non-zero and will be relevant for the vibrational effects and eventually the isotope effect on the NMR chemical shifts. Therefore, in the following both mean-square amplitudes and covariance values will be investigated, and the latter will be denoted as amplitude covariances.

In Appendix B, approximations for the expressions above within the VPT2 and DD-VPT2 frameworks will be given.

## 3. Results

### 3.1. Cyclic and Polycyclic Molecules

In this section, we discuss the vibration hole and the calculated isotopic shifts for a number of cyclic and polycyclic molecules with six to ten heavy atoms. Molecules of this kind are relatively rigid, thus, comparison with experiment is not complicated by the need of comprehensive conformation averaging. For all of these systems, the DD-VPT2 isotope shifts have been calculated and discussed in Ref. [30] for a variety of computational models. Here, we compare the DD-VPT2 values with their loc-VPT2 and LMZL counterparts and investigate the structure of the underlying vibration holes. The focus of the investigation is on three six-membered ring molecules, *viz.* pyridine, benzene, and cyclohexane (see Figure 1). For pyridine, the H/D substitution at C2 was investigated, for which recent experimental data are available [36]. For benzene, both a single H/D and a single ${}^{12}C{/}^{13}C$ substitution are investigated and compared with experimental data (see Refs. [63,64]). The ${}^{12}C{/}^{13}C$ substitution is particularly interesting since the substitution atom is bonded to more than one other atom, that is, the LZML approach is not applicable, and the loc-VPT2 approach is not obviously appropriate. For cyclohexane, the chair conformer is considered with H/D substitutions for both an axial (ax) and an equatorial (eq) H atom, and the results were compared to the experimental data from Refs. [27,65]. As a complement, we used DD-VPT2 and loc-VPT2 to determine the NMR isotopic shifts for various single H/D substitutions in norbornane and adamantane. Both of these molecules have a variety of non-equivalent NMR isotopic shifts, many of which have been acquired experimentally [66,67].

Table 1 shows the DD-VPT2 and loc-VPT2 isotopic shifts for the H/D substitution in pyridine, along with corresponding experimental values from the literature, Appendix A provide corresponding information for benzene and cyclohexane. Here and in the following, the prefix S refers to tables in the Appendix A. The LMZL and LMZL+cent values in Table 1 and Appendix A will be discussed later on. For pyridine, isotopic shifts had been acquired for a range of temperatures [36], from which values for T=0K were extrapolated [30]. Even though such a far extrapolation is inaccurate it provides a means to assess the impact of temperature effects. For benzene and cyclohexane, only the experimental values for fixed temperature were available.

Generally, the differences between corresponding DD-VPT2 and loc-VPT2 values are relatively small, the maximum deviation being 35 ppb (B3LYP) or 30 ppb (ωB97X-D) for ${}^{1}\Delta \left(C1\right)$ in benzene. There is no general trend for the loc-VPT2 values to be larger or smaller than the corresponding DD-VPT2 values (here and in the following, terms as “larger/smaller”, “increase/decrease” and so forth, always refer to absolute values.) For the two aromatic molecules, loc-VPT2 tends to predict smaller isotopic shifts than DD-VPT2, whereas the loc-VPT2 values are somewhat larger than the DD-VPT2 values for C_6_H_12_ (substitution at axial H). For C_6_H_12_ substitution at equatorial H), the two methods predict nearly the same values with differences of less than 5 ppb. Neither is there a clear trend as to which of the values are closer to experiment: for pyridine, loc-VPT2 shows slightly smaller deviations between calculation and experiment than DD-VPT2 both for 25 °C and the extrapolation to 0 K. Both loc-VPT2 and DD-VPT2 show smaller deviations from the 0 K extrapolated values than from the 25 °C values. For benzene, the loc-VPT2 values show smaller deviations from experiment (25 °C) than the DD-VPT2 values, which tend to be larger than the experimental ones. However, if one assumes that the experimental values for benzene have a similar temperature behaviour as those for pyridine the agreement of the DD-VPT2 values with experiment can be expected to be better (and that for the loc-VPT2 values poorer) for 0 K than for 25 °C. For cyclohexane (substitution at axial H), the DD-VPT2 values differ less from the 25 °C experimental values than the loc-VPT2 ones, whereas for the substitution at the equatorial H atom, both methods show about the same deviation from experiment. The B3LYP calculations deliver in most cases values that are larger than their ωB97X-D counterparts. Since ωB97X-D in turn tends to overestimate the isotope shifts, [30] B3LYP values deviate in general more from the experimental values than ωB97X-D values. In Appendix A, DD-VPT2 and loc-VPT2 isotopic shifts for norbornane and adamantane calculated with ωB97X-D are presented. A summary of these values is presented in Table 2. The data show similar trends as the ones discussed above—generally, both DD-VPT2 and loc-VPT2 values agree fairly well with experiment, the deviations between DD-VPT2 and loc-VPT2 values are generally small with no distinct trends for their differences. For norbornane, the loc-VPT2 values show a lower mean signed deviation (MSgD) than the DD-VPT2 ones. One should, however, note again that this may be a consequence of the missing temperature corrections. On the other hand, the regression coefficients between calculated and experimental values is slightly higher for DD-VPT2 than for loc-VPT2, both in the case of norbornane and of adamantane. This may indicate that DD-VPT2 reflects relevant non-local features of the vibration hole that are missing in loc-VPT2.

Altogether, for the cyclic molecules discussed here, loc-VPT2 provides similar values as DD-VPT2, which suggests that the vibration hole should indeed be localized around the C–H bond (here and in the following, the substitution site is marked by underlining. Furthermore, nearest, next-nearest, and next-next-nearest neighbors to H are denoted with no, one, and two primes, respectively). The plots in Figure 2 show the isotope effect on vibration motifs for single H/D substitutions in pyridine, benzene, and cyclohexane as well as norbornane and adamantane; specifically, the isotope effects on both the $r_{z}$ and $r_{g}$ bond lengths, the mean-square vibration amplitudes, and the amplitude covariance with $\Delta r(C\underline{H})$ are shown. The values are presented in dependence of the topological distance, that is, the number of bonds between the H atom and the respective bond (i.e., 0 for the C–H bond, 1 for geminal bonds, 2 for vicinal bonds etc.) Numerical values for selected bonds are presented in Appendix A. One sees that the vibration hole is indeed concentrated at the C–H bond—the isotope effect on *r*(CH) is about 1 mÅ ($r_{z}$) or 5 mÅ ($r_{g})$ whereas the isotope effects on *r*(CC’) are in the order of 100 μÅ for both $r_{z}$ and $r_{g}$, those on *r*(CH’) are about 300 μÅ for $r_{z}$ but less than 30 μÅ for $r_{g}$ while the values for topological distances of 2 and more generally are below 100 μÅ, the values decreasing with increasing topological distance. Generally, the isotope effects on corresponding $r_{z}$ and $r_{g}$ values are in the same order of magnitude, with the exceptions of the C–H bonds with topological distances of 1 and 4, where the $r_{g}$ values are significantly smaller than their $r_{z}$ counterparts.

The isotope effects on the mean-square amplitudes and the amplitude covariances also decay with increasing topological distances. For *r*(CH), the mean-square amplitudes are about 1500 mÅ${}^{2}$. For *r*(CC’), the isotope effect on the mean-square amplitudes are about 4 mÅ${}^{2}$ and that on the amplitude covariances about 30 mÅ${}^{2}$ for *r*(CH’), the corresponding values are about −0.3 mÅ${}^{2}$ and −6 mÅ${}^{2}$, respectively. The values decrease further monotonously with increasing topological distance. The decay is more rapid for the mean-square amplitudes than for the amplitude covariances, for which it is in turn more rapid than for the bond lengths.

For a more detailed analysis, Appendix A shows the numerical values of the ${}^{n}\Delta r$ values. The data reveal that for *r*(CH), the $r_{z}$ isotope effects are between −760 and **−2400μ**Å, whereas the $r_{g}$ values are in the interval between −5360 and −6130 μÅ. That is, the $r_{g}$ values are larger and less variant than their $r_{z}$ counterparts. Also, the values for aliphatic systems are somewhat larger than those for aromatic rings; again the difference is more distinct for $r_{z}$ than $r_{g}$. This means that the actual contraction of the C–H bond is relatively uniform, that is, the large variance the $r_{z}$ values is an artifact most relevant for the calculations. For *r*(CC’), the $r_{g}$ values are slightly larger and again somewhat less variant than the corresponding $r_{z}$ values. In contrast, the $r_{g}$ values for *r*(CH’) are considerably (a power of ten or more) smaller than their $r_{g}$ pendants; actually, the $r_{g}$ values for *r*(CH’) tend to be smaller than those for *r*(C’H”). We note that the same trends can be observed for methane (see Appendix A) even though the absolute values for the isotope effects on *r*(CH’) in methane are substantially different from those in the cyclic and polycyclic molecules.

For topological distances up to 1, the isotope effects on each of the bond lengths are quite uniform across the different compounds under consideration. For the bonds vicinal to the C–H bond, in contrast, there is a considerable variation both between different compounds and between different bonds in the same compound. In particular (see Appendix A), the isotope shifts tend to be large positive for bonds in *syn* conformation to the C–H bond, large negative for *anti* conformation, and smaller with varying sign for *gauche* conformation. We remark that the same trends are observed for the mean-square vibration amplitudes and the amplitude covariances (see Appendix A): The isotope effect on the mean-square amplitudes are in the range between −1500 and −1600 μÅ${}^{2}$ for *r*(CH), between −3.2 and −4.9 μÅ${}^{2}$ for *r*(CC’), and between −0.27 and −0.36
μÅ${}^{2}$for *r*(CH’), those on the amplitude covariances are between −26 and −36
μÅ${}^{2}$ for *r*(CC’) and between −5.9 and −8.1 for *r*(CH’) whereas the values for *r*(C’H”) and *r*(C’C”) are conformation-dependent in the same way as those for the bond lengths.

We have seen that the vibration hole is indeed localized close to the substitution site, which is in line with the rather good agreement between corresponding DD-VPT2 and loc-VPT2 isotopic shifts. One should thus expect that the LMZL approach [60], which follows a similar philosophy as the loc-VPT2 method, provides results close to the DD-VPT2 values as well. However, the MSgD of the LMZL isotope shifts for cyclohexane reported in Reference [60] differs from our DD-VPT2 values by 47 ppb (axial) and 66 ppb (equatorial), see Appendix A, these differences being too big to arise just from different basis sets and force fields or other technical details. To investigate this discrepancy, we recalculated the isotope shifts in pyridine, benzene, and cyclohexane using our own LMZL implementation. Our calculated LMZL values are systematically smaller than both their DD-VPT2 and loc-VPT2 counterparts, in line with the results form Reference [60].

This discrepancy can be traced back to the vibration holes calculated with the different methods. Table 3 shows a comparison of the DD-VPT2, loc-VPT2, and LMZL values for the isotope effects on *r*(CH) as well as the mean-square vibration amplitudes for *r*(CH) as well as the mean-square amplitudes for the azimuth angle $\varphi_{\mathrm{az}}$, and the altitude angle $\varphi_{\mathrm{alt}}$. Here, $\varphi_{\mathrm{az}}$ is defined as the angle between the C–H and the bisecting plane of the two adjacent C–H or C–N bonds, and $\varphi_{\mathrm{alt}}$, as the angle between between the C–H bond and the plane spanned by the two adjacent C–H or C–N bonds (for pyridine and benzene, $\varphi_{\mathrm{az}}$ and $\varphi_{\mathrm{alt}}$ are simply the in-plane and out-of-plane bending angles for the C–H bond, respectively.) One finds that while the DD-VPT2 absolute mean-square amplitudes for $\varphi_{\mathrm{az}}$ and $\varphi_{\mathrm{alt}}$ are significantly larger than their loc-VPT2 and LMZL counterparts, the mean-square amplitudes tend to be only slightly larger in loc-VPT2 than in DD-VPT2 and nearly identical in loc-VPT2 and LMZL. Thus, the vibration amplitudes cannot account for any major discrepancies between NMR isotopic shifts calculated by DD-VPT2, loc-VPT2, and LMZL, in particular not for the differences between loc-VPT2 and LMZL values. In contrast, the isotope effect on *r*(CH) differs significantly. LMZL employs polar coordinates centred at the C atom in the C–H bond, hence, the ${}^{n}\Delta r$ values calculated in LMZL are to be interpreted as ${}^{n}\Delta r_{g}$ values. Table 3 shows that corresponding DD-VPT2 and loc-VPT2 values for ${}^{n}\Delta r_{g}$(CH) are relatively close to each other, with the loc-VPT2 values being 0.3 to 0.4 mÅ longer than their DD-VPT2 counterparts. The LMZL values, on the other hand, are about 1.3 mÅ shorter than the corresponding loc-VPT2 values. LMZL, in contrast to DD-VPT2 and loc-VPT2, treats the vibrations along each coordinate separately, ignoring both harmonic and anharmonic couplings between the vibration modes. This description misses the centrifugal stretching of the C–H bond due to its azimuthal and altitude vibrations, and this in turn accounts for the IE on $r(C\underline{H})$ being underestimated. Given that the bond stretch is often the dominating geometry change for the isotope shift [60], this explains the tendency of LMZL to underestimate isotope shifts. This also implies that the good accuracy of the (B3LYP) LMZL values both in Reference [60] and in Reference [36], as well as the accurate values provided by our LMZL calculations with B3LYP (see Appendix A) are to be ascribed to a fortuitous error cancellation between the tendencies of B3LYP to overestimate and of LMZL to underestimate isotope shifts (as well as the temperature effects missing in the calculations).

To assess the impact of the missing anharmonic mode coupling, we redid the LMZL calculations with an additional centrifugal potential, which is derived in Appendix C. The results of these calculations, labelled LMZL+cent, are shown in Table 1 and Appendix A for the NMR isotopic shifts and in Table 3 for the vibration hole. One finds that the centrifugal corrections bring the LMZL results considerably closer to the loc-VPT2 values.

The ${}^{12}C{/}^{13}C$ substitution in benzene is particular, given that the substitution site is involved in three bonds rather than just one. Calculated and experimental NMR isotopic shifts for this substitution are presented in Table 4, isotope effects on selected geometry parameters, in Table 5. Figure 3 shows the three leading difference-dedicated vibration modes as well as the change of the $r_{z}$ geometry of benzene for both a single H/D and a single ${}^{12}C{/}^{13}C$ substitution. We remark that descriptions of all difference-dedicated vibration modes for the molecules discussed in Section 3.1 and Section 3.2 in Molden [68] format are available in the SI of Reference [30] and for the molecules discussed in Section 3.3, in the SI of the present article. The three difference-dedicated vibration modes shown for the H/D substitutions are strongly localized to the H site, similarly as for example, the corresponding mode in pyridine, which are presented in Reference [30]. In contrast, the modes for the ${}^{12}C{/}^{13}C$ are more delocalized over the molecule, which means that loc-VPT2 will not be able to describe the cahnge of the vibration amplitudes properly in this case. The situation for the change of the $r_{z}$ geometries is analogous—for a H/D substitution, the dominating change is the shortening of the C–H bond length, and DD-VPT2 and loc-VPT2 provide similar descriptions (see Figure 3a). For a ${}^{12}C{/}^{13}C$ substitution, in contrast, loc-VPT2 cannot properly describe the shortening of the C–H bond (see Figure 3b): since the position of the H atom is fixed this bond can not get shorter in terms of $r_{z}$ values as it should but gets even a bit longer (see Table 5). However, the IE on $r_{g}$(CH) becomes negative as it should be, and the value is close to its DD-VPT2 pendant. Obviously, the amplitudes of the lateral vibrations manage to “cover up” to some extent for the missing geometric flexibility in loc-VPT2. As a consequence, the loc-VPT2 isotope shifts (see Table 4) are indeed smaller than the corresponding DD-VPT2 values but the differences are moderate. The MSgD and root-of-mean-square (RMS) deviations are even smaller for loc-VPT2 than for DD-VPT2 (again, missing temperature effects may affect the results). However, loc-VPT2 incorrectly predicts ${}^{3}\Delta \left(H3\right)$ to be larger than ${}^{4}\Delta \left(H4\right)$, in distinction to DD-VPT2.

Altogether, in cases where the substitution site has only one bond to the rest of the system, *a priori* local methods may in principle provide a reasonable description of the vibration hole and, in extension, reasonable values for the NMR isotope shifts. The major deviations between LMZL and DD-VPT2 results are caused by the missing centrifugal mode coupling, not by the local approach. Still, the use of DD-VPT2 is to be preferred since the latter at comparable computational expenses calculates the vibration hole without *ad-hoc* assumptions and is superior in reflecting trends in NMR isotope shifts, as we have seen for the ${}^{12}C{/}^{13}C$ substitution in benzene and will get corroborated in the following sections.

### 3.2. Large Systems: Halonium Complexes of Bis(Phenylethynyl)Benzene Derivatives

Erdélyi et al. [37,39,41] have investigated halonium complexes of 1,2-bis(pyridin-2-ylethynyl)-benzene (**bpeb**) derivatives (Figure 4) where the 2-pyridyl moieties were substituted in their *para*-positions with several electron-donating and -withdrawing groups. The [NXN]^+^ moieties in these complexes are potential building blocks in supramolecular frameworks [69,70] and as parts of halonium-donating and oxidizing agents [41,71] (for a more comprehensive bibliography, see Reference [41]. A central question in these investigations was whether the [NXN]^+^ moiety in these compounds is statically symmetric or rapidly interconverts between [N−X···N]^+^ and [N···X−N]^+^ tautomers, in particular whether the substituent groups influence this symmetry behaviour. This issue was investigated with a range of experimental methods, among them isotopic perturbation of the equilibrium (IPE) [32], which involved measuring the isotopic shifts at the nuclei C2 through C6 upon a H/D substitution at H2 over a range of temperatures. It turned out that all complexes are statically symmetric, that is, all measured isotopic shifts are purely intrinsic in origin; moreover, the NMR isotopic shifts could be extrapolated to 0 K in the same way as for pyridine in Section 3.1. The **bpeb** complexes are not only larger than the molecules studied in Section 3.1, with topological distances from the C–H bond up to 9, but also somewhat more flexible while they at the same time still have one distinct ground-state conformation (i.e., we do not need to perform conformation averaging). It is thus interesting whether the correlation holes in the **bpeb**-derivative complexes behaves similarly as those in the cyclic and polycyclic molecules studied in Section 3.1, in particular pyridine. It should be noted that the calculations in this subsection have been done with a different computational model than those in Section 3.1 (see Section 4 for details), hence the results for pyridine presented in the two sections are slightly different.

Figure 5 shows the DD-VPT2 and loc-VPT2 NMR isotopic shifts as compared to experiment (0 K), the numerical values are found in Appendix A. A summary of the results is presented in Table 2. Similarly as in Section 3.1, the differences between DD-VPT2 and loc-VPT2 results are relatively small, and there is no clear superiority for any of them. Both the DD-VPT2 and the loc-VPT2 values reflect the substituent effects on ${}^{1}\Delta$(C2) reasonably well but, as with norbornane and adamantane, the regression coefficient between calculated and experimental values is slightly higher for DD-VPT2 than for loc-VPT2.

Figure 6 shows a synopsis of the vibration holes in the **bpeb** derivatives in a format analogous to Figure 2, except that the $r_{z}$ geometries are omitted in Figure 6 to avoid too congested representations. The corresponding geometry parameters for pyridine are marked with shaded black ticks to facilitate a direct comparison between pyridine and the **bpeb**-derivative complexes. Just as for the cyclic molecules discussed in Section 3.1, both the changes of the geometry and the changes in mean-square amplitudes and amplitude covariances are dominated by the contributions from the C–H bond. Also, the decay of the geometry parameters in dependence of the topological distance is strongest for the mean-square amplitude, next strongest for the amplitude covariances, and weakest for changes of the $r_{g}$ geometries. In particular, for the geometry changes, no further decay is found for a topological distance above 3 to 4. Similarly, the vibration amplitudes and amplitude covariances for the C–H bonds do not decay any longer for topological distances beyond 6. Only for the mean-square amplitudes and amplitude covariances of the C–C bonds does one find a consistent decay over the whole range of topological distances. It should, however, be mentioned that the isotope effects on both the $r_{g}$ bond lengths (about or below 100 μÅ), vibration amplitudes (about 10${}^{-4}$
μÅ) and amplitude covariances (about 10${}^{-1}$
μÅ) are rather small, such that the long-range tail of the vibration hole makes only a small contribution to the NMR isotopic shifts. Also, one finds that the vibration hole in pyridine is similar to that in the isotope-substituted pyridyl moiety of the **bpeb**-derivative complexes; corresponding parameters have comparable values. For a more detailed discussion, the geometry parameters for topological distances up to 2 are provided in Appendix A. The values reveal that the isotope effect on both $r_{g}$(CH) and the corresponding vibration amplitude are quite uniform across the set of **bpeb**-derivative complexes as well as pyridine. One finds ${}^{n}\Delta r_{g}$$\left(C\underline{H}\right)=-5558\;\mu \overset{˚}{A}$ for pyridine and values between −5521 and $-5586\;\mu \overset{˚}{A}$ for **bpeb–1a** to **bpeb-2**. The standard deviation for the ${}^{n}\Delta r_{g}$$\left(C\underline{H}\right)$ values in the set **bpeb-1** to **bpeb-5** is just 28 μÅ, the average difference of ${}^{n}\Delta r_{g}$(CH) between pyridine and the **bpeb-derivative** complexes is just 2 μÅ. Also, there is no clear correlation between ${}^{n}\Delta r_{g}$$\left(C\underline{H}\right)$ and ${}^{1}\Delta$(C2). The mean-square amplitude for *r*(CH) is slightly larger in pyridine (−1517 mÅ${}^{2}$) as compared to the **bpeb**-derivative complexes (−1496 to $-1500m{\overset{˚}{A}}^{2}$). For the CC’ and CN’ bonds (topological distance of 1), the deviation between the value for pyridine on the one hand and the average values for the **bpeb**-derivative complexes on the other hand is actually larger [${}^{n}\Delta r_{g}$(CC’): 33 μÅ, ${}^{n}\Delta r_{g}$(CN’): 98 μÅ], that is, there are actually slight differences in the geometry of the pyridyl moiety between pyridine and the **bpeb**-derivative complexes. Generally, the variation of a given geometry parameter across the set of compounds tends to be smaller for **bpeb**-derivative complexes than for the cyclic molecules studied in Section 3.1, which can be explained with the smaller conformational diversity in the former. Furthermore, loc-VPT2 and DD-VPT2 give quite similar results for the isotope effects on the geometry parameters, and in line with the findings from Section 3.1, the loc-VPT2 values tend to be slightly larger than the DD-VPT2 ones.

The isotope effect on the $r_{z}$ geometries can be visualized in a similar way as shown above for benzene (see Figure 3). Figure 7 shows the change of the geometry for pyridine as well as **bpeb-1a** and **bpeb-1b** for both DD-VPT2 and loc-VPT2. Even though the largest changes in the geometry are found around the substitution site one finds that the geometry of the pyridyl moiety is somewhat different in the three cases. In addition, for the two **bpeb**-derivative complexes, there are global changes in the geometry, which are facilitated by the flexibility of the ethynyl [37] and [NXN]${}^{+}$ moieties. These changes are substituent-dependent, one finds that they are considerably more distinct in **bpeb-1d** than in **bpeb-1a**. One has, however, to bear in mind that the geometry changes are exaggerated by a factor of 400 in Figure 7, that is, the actual geometry changes are rather small both in **bpeb-1a** and **bpeb-1d**. Altogether, in spite of some minor peculiarities the behaviour of the vibration hole in the **bpeb**-derivative complexes largely resembles that for the molecules discussed in Section 3.1.

### 3.3. Molecules with Intramolecular Hydrogen Bonds: Salicyl Aldehyde Derivatives

In the sets of molecules investigated in Section 3.1 and Section 3.2, we found the vibration holes to be well localized around the substitution site. For all these molecules (except the case of an ${}^{12}C{/}^{13}C$ substitution in benzene), the substitution site was involved in one covalent bond only and not part of for example, an intra-molecular hydrogen bond. However, NMR isotopic are potentially valuable probes for the investigation of inter- and above all intramolecular weak interactions, since they reflect the vibration pattern motif and, in extension, the molecular potential around the substitution site. Several authors, among others Limbach and co-workers (see e.g., References [17,18,19,59]) and Hansen and co-workers (see e.g., References [14,15,20,58,72,73]) have studied NMR isotopic shifts in hydrogen-bonded systems. It is thus interesting to know what the vibration hole looks if the atom at the substitution site is part of an intramolecular hydrogen bond. Recently, Hansen et al. [20] investigated a number of conventional as well as sterically compressed salicyl aldehyde (**sal**) derivatives as shown in Figure 8 (our numbering is consistent with that in Reference [20]; our stem compound **sal** is compound **13** there). In distinction to the set of **bpeb**-derivative complexes studied in Section 3.2, the **sal** derivatives in Reference [20] show a much more diversified range of substituents, comprising electron-donating and -withdrawing groups as well as bulky groups that sterically constrain the HC=O···OH moiety and additional hydroxyl groups that can involve in (weak) hydrogen bonds. Also, the substituents in the **sal** derivatives are not restricted to one position as in the **bpeb**-derivative complexes. The study in Reference [20] was aimed at finding theoretical and experimental indicators for the strength of the intra-molecular hydrogen bonds. Among other things, the ${}^{2}\Delta$(C2) values of the **sal** derivatives were acquired for that purpose, which proved to vary by several 100 ppb as compared to just 30 ppb for the experimental ${}^{1}\Delta \left(C2\right)$ isotopic shifts in **bpeb-1a** to **bpeb-2**. The strong substituent dependence of the ${}^{2}\Delta$(C2) in **sal** to **sal-12** makes this set of compounds an interesting testbed for the theoretical prediction of NMR isotopic shifts as well for the investigation of the vibration hole.

Figure 9 shows an overview over the DD-VPT2 and loc-VPT2 values for ${}^{2}\Delta$(C2) in **sal** to **sal-12** versus their experimental counterparts (Figure 9a) and the DD-VPT2 versus the loc-VPT2 values (Figure 9b). The loc-VPT2 values vary considerably less across the set of compounds (standard deviation 73 ppb) than both the experimental (188 ppb) and DD-VPT2 (203 ppb) values. All loc-VPT2 values are too positive whereas some of the DD-VPT2 values are too positive and some too negative. On the other hand, loc-VPT2 correctly predicts all ${}^{2}\Delta \left(C2\right)$ values to be negative whereas DD-VPT2 predicts a positive value (73 ppb) for **sal-12**. Figure 9b shows a clear correlation between DD-VPT2 and loc-VPT2 values ($r^{2}=0.95$) but the slope of the regression curve is just 0.35. Altogether, these findings suggest that loc-VPT2 misses important features of the vibration hole and its variation between the different **sal** derivatives and, as a consequence, underestimates the variance of the ${}^{2}\Delta$(C2) values. On the other hand, the DD-VPT2 values are not consistently superior to their loc-VPT2 counterparts—the RMS deviation between calculated and experimental values is only moderately larger for DD-VPT2 (164 ppb) than for loc-VPT2 (201 ppb), and the correlation coefficient is even lower for DD-VPT2 (0.641) than for loc-VPT2 (0.771). That is, neither loc-VPT2 and DD-VPT2 provides a quantitative prediction of the ${}^{2}\Delta$(C2) values. To get a better understanding of these shortcomings we do a more detailed analysis of the correlation hole as described with the two methods. Unlike in Section 3.1 and Section 3.2, our focus is in this case not on the long-range behaviour of the vibration hole but its behaviour in the C=O···HO moiety.

We remark that the mean-square amplitudes and amplitude covariances for the geometry parameters of the HC=O···OH moiety (see Appendix A) as well as the isotope effects on these quantities are rather uniform across the set **sal** to **sal-12**. This implies that the variation between the NMR isotopic shifts between the compounds should be due to changes in the geometry. Besides, the variation in the $r_{z}$ and $r_{g}$ values for $r(O–\underline{H})$ and $r(O\cdots \underline{H})$ are largely parallel—the slope for the regression line $r_{g}$ vs. $r_{z}$ is 0.882 for $r(O–\underline{H})$ and 1.0003 for $r(O\cdots \underline{H})$, the regression coefficients are 0.9888 and 0.9973, respectively. For r(O,O), in particular, the isotope effect on the mean-square amplitude is small (<$100\;m{\overset{˚}{A}}^{2})$, and the ${}^{n}\Delta r_{z}$ and ${}^{n}\Delta r_{g}$ values are nearly identical. Also, the amplitude covariance for r(O,O) and its isotope effect are small, that is, each atom in the HC=O···OH moiety moves essentially in the average molecular potential of the other ones, and the response of r(O,O) to the position of H is described most appropriately in terms of the $r_{z}$ geometry.

Figure 10a shows $r_{e}\left(O–\underline{H}\right)$ versus $r_{e}\left(O,O\right)$ for the intra-molecular hydrogen bonds, Figure 10b the corresponding ${}^{n}\Delta_{z}$ values. The variance both in the $r_{e}$ and the ${}^{n}\Delta r_{z}$ values is considerably larger than for the **bpeb**-derivative complexes: The standard deviation of the $r_{e}\left(O–\underline{H}\right)$ values is 6.7 mÅ, as compared to 0.2 mÅ for $r_{e}\left(C\underline{H}\right)$ in **bpeb-1a** to **bpeb-2**. For ${}^{n}\Delta r_{z}\left(O–\underline{H}\right)$, the standard deviation is 1.2 mÅ, as compared to 28 μÅ for ${}^{n}\Delta r_{z}\left(C\underline{H}\right)$ in **bpeb-1a** to **bpeb-2**. Figure 10a shows a negative correlation between $r_{e}\left(O–\underline{H}\right)$ and $r_{e}\left(O,O\right)$, in line with the fact that the O–H bond gets stretched as the hydrogen bond gets stronger (as is the case when the donor and acceptor atoms come closer to each other). The corresponding isotope effects are shown in Figure 10b. All ${}^{n}\Delta r_{z}\left(O–\underline{H}\right)$ except that for **sal-12** are negative. (The corresponding ${}^{n}\Delta r_{g}$ values, including that for **sal-12**, are all negative, as is expected for a substitution with a heavier isotope, see Appendix A.) Again, there is a negative correlation between ${}^{n}\Delta r_{z}\left(O–\underline{H}\right)$ and ${}^{n}\Delta r_{z}\left(O,O\right)$, which can be rationalized in the same way as for the $r_{e}$ values. There are a number of outliers, above all **sal-12** and in second instance **sal-5**, **sal-9**, and **sal-10**, for which ${}^{n}\Delta r_{z}\left(O,O\right)$ is less positive than expected. These compounds are sterically constrained, which counteracts the expansion of the O,O distance. We also note that for compounds **sal-1**, **sal-4**, and **sal-7**, ${}^{n}\Delta r_{z}\left(O,O\right)<0$ (i.e., the hydrogen bond tends to get stronger) even though ${}^{n}\Delta r_{z}\left(O–\underline{H}\right)$ is negative (i.e., the H atom moves away from the acceptor O atom) and that ${}^{n}\Delta r_{z}\left(O,O\right)$ is large negative for **sal-12** even though ${}^{n}\Delta r_{z}\left(O–\underline{H}\right)$ is quite small, which appears to contradict the explanation above.

This apparent contradiction can be resolved by an analysis of ${}^{n}\Delta r_{z}$ for the complete molecule. Figure 11 shows the three leading difference-dedicated vibration modes and the isotope effect on the $r_{z}$ geometries for the stem compound **sal** as well as for **sal-2**, which has the largest positive ${}^{n}\Delta r_{z}$(O,O) and **sal-12**, which shows the largest negative ${}^{n}\Delta r_{z}$(O,O) among all compounds investigated. The difference-dedicated vibration modes are well localized at the H atom, and the isotope effects on the vibration amplitudes are rather uniform for the three compounds. The isotope effects on the $r_{z}$ geometries, however, vary markedly—while for **sal** the vibration hole is largely concentrated on the OH bond, it is delocalised over the whole C=O···HO moiety in **sal-2** and **sal-12** (as, to a lesser extent, for the other **sal** derivatives). We point out that in Figure 11, geometry changes are enhanced just by a factor of 80, as compared to factors of 400 to 10,000 in Figure 3 and Figure 7. The comparison between DD-VPT2 and loc-VPT2 geometries corroborates that the latter, as any *a priori* local method for these molecules, misses important features of the vibration hole and cannot be expected to provide accurate NMR isotope shifts. Also, one finds that both the whole hydroxyl group and the whole aldehyde group are involved in the geometry changes. Thus, a suitable calculation method needs to treat at least the C=O···HO moiety as flexible. A closer inspection of the geometry changes reveals that the isotopic substitution changes not only the O–H bond length but also the bond angle $\varphi (C2–O–\underline{H})$. In **sal-2**, $\varphi (C2–O–\underline{H})$ opens upon isotopic substitution, which increases $r(O\cdots \underline{H})$ and weakens the hydrogen bond. In **sal-12** (and some other compounds, including **sal-1**, **sal-4**, and **sal-7**), $\varphi (C2–O–\underline{H})$ closes instead, which eventually decreases r(O,O). Also, one finds that the changes both in $\Delta r(O–\underline{H})$ and in $\varphi (C2–O–\underline{H})$ are roughly the same for loc-VPT2 as for DD-VPT2. The data for ${}^{n}\Delta_{z}r\left(O–\underline{H}\right)$ and ${}^{n}\Delta_{z}\varphi \left(C2–O–\underline{H}\right)$ (see Appendix A) show that this holds for all compounds: the slope for the loc-VPT2 vs. DD-VPT2 values is 0.722 for ${}^{n}\Delta_{z}r\left(O–\underline{H}\right)$ and 0.860 for ${}^{n}\Delta_{z}\varphi \left(C2–O–\underline{H}\right)$, the regression coefficients are 0.9885 and 0.9872, respectively. In other words, the variation of ${}^{n}\Delta_{z}r\left(O–\underline{H}\right)$ and ${}^{n}\Delta_{z}\varphi \left(C2–O–\underline{H}\right)$ across the compounds is about 30% and 15%, respectively larger in DD-VPT2 than in loc-VPT2.

While the loc-VPT2 calculations cannot provide a correct description of the vibration hole they are, in connection with the DD-VPT2 results, helpful to elucidate the mechanism behind the NMR isotopic shifts: The findings in the previous paragraph imply that the isotope effects on $\Delta r(O–\underline{H})$ and in $\varphi (C2–O–\underline{H})$ are in first instance a direct response to the anharmonic molecular potential around the site of the H atom. The change in r(O,O) is then an indirect response to the displacement of the H atom, and there is only a relatively weak feedback from the displacement of the acceptor O atom to the geometry around the O–H bond. This picture is supported by Figure 12a, which shows ${}^{n}\Delta_{z}r\left(O,O\right)$ in dependence of ${}^{n}\Delta_{z}\varphi \left(C2–O–\underline{H}\right)$. For the majority of the compounds, one finds a linear relationship between the two isotope effects, with the same outliers as for the relationship between ${}^{n}\Delta_{z}r\left(O–\underline{H}\right)$ and ${}^{n}\Delta_{z}r\left(O,O\right)$ shown in Figure 10b. The changes in r(O,O) leverage the isotope effect on ${}^{2}\Delta \left(C2\right)$ beyond the direct effect from the displacement of the H atom. This leverage gives rise to the relatively large variation of the ${}^{2}\Delta \left(C2\right)$ values observed. DD-VPT2 reflects this leverage effect, and the standard deviation of the DD-VPT2 values for ${}^{2}\Delta$(C2) is about the same as for the experimental values. In contrast, loc-VPT2 misses this leverage and predicts a set of ${}^{2}\Delta$(C2) values with too small a standard deviation. A similar error has been observed in the multi-component DFT calculations for 2-hydroxy-acetophenone derivatives by Udagawa et al. [44] (see Figure 3 in that reference) even though those calculations do account for geometry changes around the intra-molecular hydrogen bond. This leverage also explains why the loc-VPT2 results correlate quite strongly with their DD-VPT2 counterparts (see Figure 9b). On the other hand, if the description of the molecular potential around the H atom is inaccurate, then these inaccuracies will be leveraged in the DD-VPT2 calculations in the same way. This may account for the large deviations between DD-VPT2 calculations and experiment for some of the molecules. Specifically, the potential around an atom in an intra-molecular hydrogen bond may be strongly anharmonic, and the VPT2 approach with just cubic corrections to the harmonic force field may be insufficient for an appropriate description. We consider this limitation of the molecular potential to be a probable cause for the shortcomings of the DD-VPT2 calculations. To test this hypothesis we studied the correlations between the deviation ${}^{2}\Delta {\left(C2\right)}_{\mathrm{DD}-\mathrm{VPT}2}-{}^{2}\Delta {\left(C2\right)}_{\exp}$ and different geometry parameters. While there was no perfect correlation for any geometry parameter we found a regularity in the relationship between ${}^{n}\Delta r_{z}\left(O,O\right)$ and the deviation (see Figure 12b)—the majority of the data points lie close to a straight line. There are a number of outliers (**sal-11**, **sal-5**, **sal–10**), in second instance **sal-3a,b**); all these outliers are on the same side of the mentioned straight line. A similar behaviour is found for ${}^{n}\Delta r_{z}(O\cdots \underline{H})$ and ${}^{n}\Delta_{z}\varphi \left(C2–O–\underline{H}\right)$, which both in turn are strongly correlated with ${}^{n}\Delta r_{z}\left(O,O\right)$, no similar regularity was found for any other geometry parameter. While the relationship in Figure 12b is no distinct proof it is in line with our hypothesis, and future investigations of the **sal** derivatives will focus at a more appropriate description of the molecular potential around the substitution site.

## 4. Implementation, Computational Details

Standard VPT2 and DD-VPT2 calculations were performed with the help of the Python package described in more detail in Reference [30], LMZL calculations with the Perl/Fortran framework described in the same reference. We note that our LMZL implementation, in distinction from the original one [60], determines the changes in coordinates and the vibration amplitudes perturbatively, in order to make the LMZL results better comparable to the DD-VPT2 and loc-VPT2 results. All underlying quantum-chemical calculations were performed with the Gaussian 09 program package [74]. Anharmonic force constants and gradients of chemical shielding constants were calculated by numerical differentiations with the step widths determined in Reference [30], that is, $h_{\mathrm{FF}}=0.005$ a.u. for the anharmonic force fields and $h_{\mathrm{NMR}}=0.05$ a.u. for the NMR gradients. The anharmonic force field constants were calculated as first-order numerical derivatives of analytical second-order energy gradients (Δk approach in the nomenclature of Reference [30] for benzene and all molecules in Section 3.3, otherwise as second-order numerical derivatives of analytic forces ($\Delta^{2}F$ approach).

Density-functional theory (DFT) was used for all calculations, employing the ωB97X-D [75] and in some cases the B3LYP [76,77,78,79,80] exchange and correlation (XC) functionals and Jensen’s polarization-consistent basis sets [81,82]. In selected calculations, solvent effects were accounted for by the polarized continuum model (PCM) [83]. Ultra-fine grids (Gaussian keyword Grid=UltraFine) were used for all numerical XC integrations.

The calculations in Section 3.1 were performed with the pc-2 basis set [81,84] for force-field calculations and the pcS-2 basis set [82] for the calculation of chemical shielding constants. Solvent effects were incorporated in the NMR calculations for pyridine (dichloromethane as solvent) and benzene (acetone). The solvent corrections were calculated for ωB97X-D only and were used for both ωB97X-D and B3LYP calculations.

For the calculations in Section 3.2, the ωB97X-D/basis2//ωB97X-D/basis1 level of theory was used where basis1 [basis 2] are mixed-level basis sets consisting of aug-pc-1 [aug-pcS-2] for the N atoms in the [N–X–N]${}^{+}$moiety, pc-1 [6-311G(d)] for Br [85], 6-311G(d) [6-311G(d)] for I [86], and pc-1 [pcS-2] otherwise. Solvent effects for dichloromethane were taken into account both in the the geometry optimizations and force-field calculations and in the NMR calculations. The force fields were rigidified in the same way as described in Reference [30] to suppress the free rotations of the methyl groups in **bpeb-1b**.

The calculations in Section 3.3 were performed at the ωB97X-D/basis4//ωB97X-D/basis3 level of theory was used where basis3 is a mixed-level basis set constructed from aug-pc-2 for the O⋯H–O moiety, pc-2 for the atoms C1, C2, and C7 and pc-1 otherwise. Basis4 consists of aug-pcS-2 for the H–C=O⋯H–O moiety and pcS-2 for all other atoms. Note that, as a consequence of the mixed-level basis set used, **sal-3a** and **sal-3b** are predicted to have slightly different geometries and ground-state energies in spite of their having the same structure. Solvent effects for trichloromethane were taken into account in the NMR calculations.

For the calculations in Section 3.3, a cut-off value of $10^{-2}$ for $\left|\kappa_{I}\right|$ was used, for the ${}^{12}C{/}^{13}C$ substitution in benzene, $10^{-4}$, in all other cases, $10^{-3}$. The latter two values were chosen for consistency with Reference [30].

Basis sets not available in the respective quantum-chemistry packages were retrieved from the EMSL Basis-set exchange database [87,88].

Molecular graphics were prepared with the Jmol viewer [89].

## 5. Conclusions

We have used the recently developed DD-VPT2 approach [30] to study the structure of the vibration hole for isotopic substitutions in different groups of compounds and discuss the performance and limitations of *a priori* local methods for the calculation of NMR isotopic shifts. A number of conclusions can be drawn from this work:1For a H/D substitution on a C-bonded H atom (thus not involved in a hydrogen bond), the vibration hole is distinctly localized at the C–H bond, with respect to both bond lengths and mean-square amplitudes as well as amplitude covariances. Regarding the bond-length contractions, this localization is seen more clearly in the $r_{g}$ than in the $r_{z}$ values. The ${}^{n}\Delta r_{g}$ values for the C–H bonds are typically around $-5\;m\overset{˚}{A}$, those for the C–C’ bonds around −150 to $-250\;\mu \overset{˚}{A},$ the remaining ones generally between 0 and $-100\;\mu \overset{˚}{A}$. In particular, the ${}^{n}\Delta r_{g}$ values for the C–H’ bonds are very small, between −20 and $20\;\mu \overset{˚}{A}$. The isotope effect on the mean-square amplitude for the C–H bonds is around $-1500\;m{\overset{˚}{A}}^{2}$, −2.5 to $-5\;m{\overset{˚}{A}}^{2}$ for the C–C’ bonds, between 0 and $-2.5\;m{\overset{˚}{A}}^{2}$ for the C’–C” bonds, and between 0 and $-0.4\;m{\overset{˚}{A}}^{2}$ for the remaining bonds. For the amplitude covariances with $r(C–\underline{H})$, the corresponding values are −25 to $-36\;m{\overset{˚}{A}}^{2}$ (C–C’ bond), −5 to $-8\;m{\overset{˚}{A}}^{2}$ (C–H’ bond), −7 to $8\;m{\overset{˚}{A}}^{2}$ (C’–C” bond) and −2 to $2\;m{\overset{˚}{A}}^{2}$ otherwise. One sees that the localization is (expectedly) stronger for the mean-square amplitudes than for the amplitude covariances.2The isotope effects are quite uniform across the different compounds for the C–H, C–C’, and C–H’ bonds. For vicinal bonds in aliphatic molecules, the isotope effects distinctly depend on the conformation relative to the C–H bond.3Both for the cyclic and polycyclic systems from Section 3.1 and the **bpeb**-derivative complexes from Section 3.2, the parameters describing the correlation hole decay from the substitution site up to a topological distance of 3 to 4. Beyond that value, no consistent decay of the values is found (see in particular Figure 6). However, the geometry and amplitude changes beyond this topological distance are rather small (absolute values up to $0.1\;\mu \overset{˚}{A}$ or $1\;m{\overset{˚}{A}}^{2}$, respectively) so that they are expected to make only minor contributions to the NMR isotopic shifts.We note though that the large molecules studied in this work all are aromatic with a contiguous set of π orbitals over the whole molecule. The vibration hole may behave differently in an aliphatic molecule of comparable size.4The loc-VPT2 description of the vibration hole at the C–H bond follows the corresponding DD-VT2 description relatively closely. Generally, loc-VPT2 predicts slightly larger isotope effects on geometry parameters than DD-VPT2.5As a consequence of 4, the loc-VPT2 values for the NMR isotopic shifts are relatively close to their DD-VPT2 counterparts, and there are no systematic trends in the deviations between DD-VPT2 and loc-VPT2 values.Still, we recommend to use the DD-VPT2 approach since it predicts NMR isotopic shifts at a cost comparable to loc-VPT2 but without *a priori* assumptions on the vibration hole. Furthermore, the correlation between calculated and measured values tends to be slightly higher for DD-VPT2 than for loc-VPT2, which indicates that DD-VPT2 reflects relevant features of the vibration hole that are missing in loc-VPT2.6The LMZL approach by Yang and Hudson [60] systematically underestimates the isotope effect on $r(C–\underline{H})$ and, in extension, the NMR isotopic shifts. This is due not to the *a priori* local approach but to the missing anharmonic coupling between bond-stretching and bond-bending as well as out-of-plane bending vibrations. With the help of a relatively simple centrifugal correction (see Appendix C), this shortcoming can be remedied effectively.7For a ${}^{12}C{/}^{13}C$ substitution in benzene (i.e., the substitution site is involved in more than one bond), loc-VPT2 can in principle not provide a correct description of the vibration hole. Still, the isotope effects on the $r_{g}$ bond lengths are relatively close to their DD-VPT2 counterparts, and the deviations between the DD-VPT2 and loc-VPT2 NMR isotopic shifts are moderate. To some extent, the changes in the vibrational amplitudes “cover up” for the lacking flexibility of the loc-VPT2 approach.8If the H atom at the substitution site is involved in an intra-molecular hydrogen bond the picture changes thoroughly. In this case, the vibration hole is delocalized all over the hydrogen-bond moiety; in the case of the **sal** derivatives studied in Section 3.3, over the whole HC=O···OH moiety. As a consequence, loc-VPT2 calculations cannot provide a proper description of the vibration hole and not either correctly predict NMR isotopic shifts. However, loc-VPT2 calculations can be used, in conjunction with DD-VPT2 calculations, to elucidate the mechanism behind the NMR isotopic shifts.9For the **sal** derivatives studied in Section 3.3, the substituent effects on ${}^{2}\Delta \left(C2\right)$ stretch over a range of several 100 ppb. loc-VPT2 calculations predict values with too low a variance, since loc-VPT2 misses important features of the vibration hole. The variance of the DD-VPT2 values is approximately correct; however, neither with regard to the RMS deviation or the correlation between calculated and experimental ${}^{2}\Delta \left(C2\right)$ values is DD-VPT2 clearly superior to loc-VPT2. There is a clear correlation between loc-VPT2 and DD-VPT2 values.10An analysis of the vibration holes for the **sal** derivatives calculated with DD-VPT2 and loc-VPT2 reveals that the two descriptions agree relatively well at the O–H bond. This gives at hand that these changes are mainly a direct response to the anharmonic molecular potential of at the H site, whereas the geometry change in the HC=O moiety is a response to the displacement of the H atom. This is corroborated by the (albeit not perfect) correlations between ${}^{n}\Delta r_{z}\left(O–\underline{H}\right)$ and ${}^{n}\Delta \varphi_{z}\left(C–O–\underline{H}\right)$ on the one hand and ${}^{n}\Delta r_{z}\left(O,O\right)$ on the other hand.11Point 10 gives at hand that a part of ${}^{2}\Delta \left(C2\right)$ is caused directly by the change of the geometry at the O–H bond (this part is covered by loc-VPT2), while the geometric response of the HC=O moiety (only covered in DD-VPT2) “leverages” and amplifies this effect. This explains the correlation between loc-VPT2 and DD-VPT2 ${}^{2}\Delta \left(C2\right)$ values and offers an explanation for the limited accuracy of the former—the potential around the H site is supposed to be strongly anharmonic, and VPT2 may not be sufficient for a proper description. The errors in this description are then “leveraged” by the HC=O moiety, which causes the relatively large errors in the DD-VPT2 ${}^{2}\Delta \left(C2\right)$ values.

The conclusions regarding the **sal** derivatives give at hand that a proper prediction of NMR isotopic shifts around intra-molecular hydrogen bonds is impossible with *a priori* local methods but that DD-VPT2 may be insufficient as well because of the limitations in the VPT2 treatment of anharmonicities. To remedy this shortcoming, one needs thus to go beyond the VPT2 level. One possible approach is to incorporate higher anharmonicities, at least the quartic term of the molecular potential. However, the possible shapes of this potential may vary greatly between different classes of compounds; for instance, double-well potentials may occur. Thus, a better approach is to treat the molecular potential non-perturbatively. Christiansen and co-workers [90,91,92,93] have developed methods to construct anharmonic potentials for polyatomic systems in an incremental fashion. However, for a molecule with 100 or more normal modes, these methods are not feasible. An alternative approach would be to treat only selected coordinates non-perturbatively while keeping the VPT2 description for the rest of the molecule. A refinement along these lines would make the DD-VPT2 approach more readily applicable to an interesting class of molecules.

## Figures and Tables

**Figure 1 molecules-25-02915-f001:**
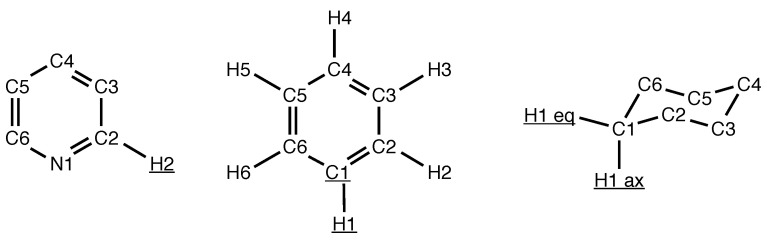
The numbering of atoms for pyridine, benzene, and cyclohexane. The substitution sites are underlined.

**Figure 2 molecules-25-02915-f002:**
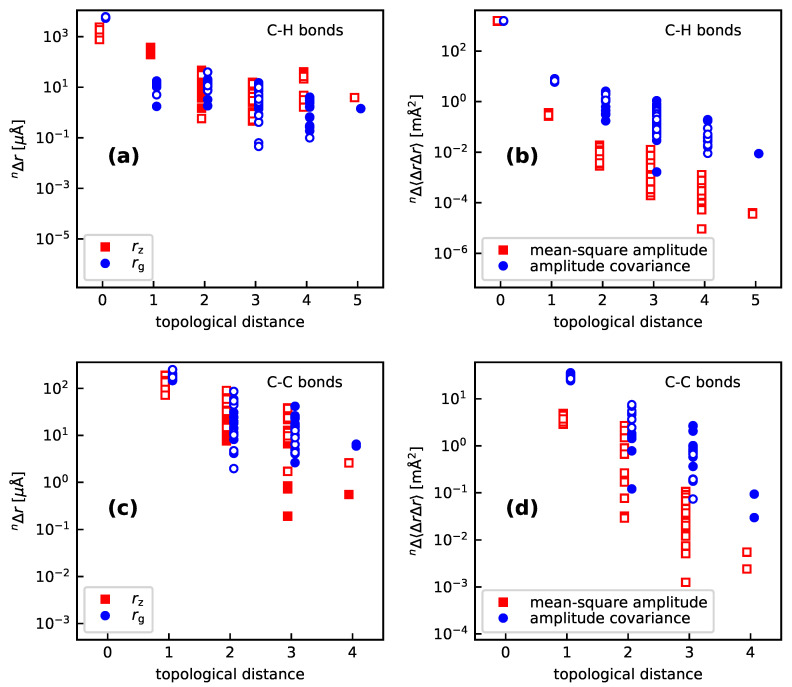
Isotope effects on the C–H (**a**) and C–C (**c**) bond lengths, as well as the mean-square vibration amplitudes and the covariance of the amplitudes with ${}^{n}\Delta r\left(C\underline{H}\right)$ for C–H (**b**) and C–C (**d**) bonds, for all H/D isotopic substitutions in the molecules investigated in Section 3.1. The topological distance is the number of bonds between the present bond and the H site, that is, 0 for the C–H bond itself, 1 for geminal bonds, 2 for vicinal bonds and so forth. The graphs show the absolute values of the isotope effects, full and hollow markers indicate positive and negative values, respectively. Calculations done at the ωB97X-D//pc-2 level of theory.

**Figure 3 molecules-25-02915-f003:**
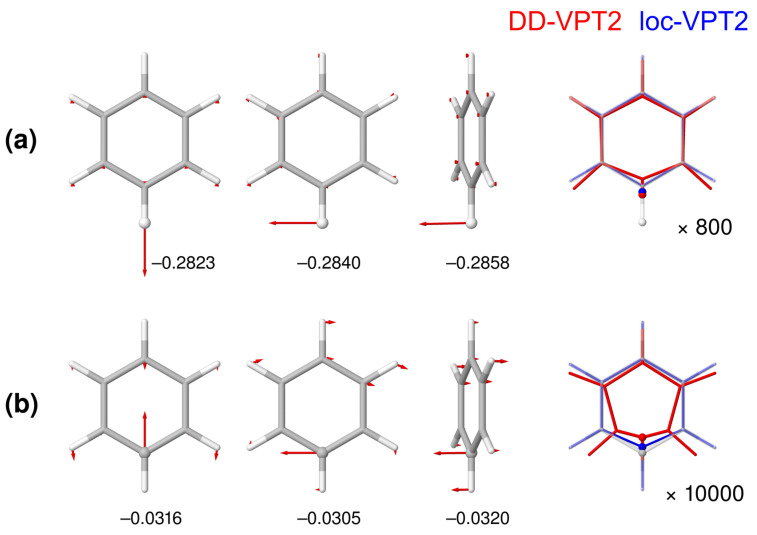
The three leading difference-dedicated vibration modes (first three columns) and isotope effects on the $r_{z}$ geometry (rightmost column) of benzene for (**a**) a single H/D substitution and (**b**) a single ${}^{12}C{/}^{13}C$ substitution. The substitution sites are marked by atomic spheres. For the difference-dedicated vibration modes, the numbers below the structures give the corresponding weight factors $\kappa_{i}$. In the rightmost column, semitransparent structures show the $r_{e}$ geometry, red and blue structures show the geometry changes as calculated with DD-VPT2 and loc-VPT2, respectively. Geometry changes enhanced by a factor of 800 for (**a**) and a factor of 10,000 for (**b**). Calculations done at the ωB97X-D//pc-2 level of theory.

**Figure 4 molecules-25-02915-f004:**
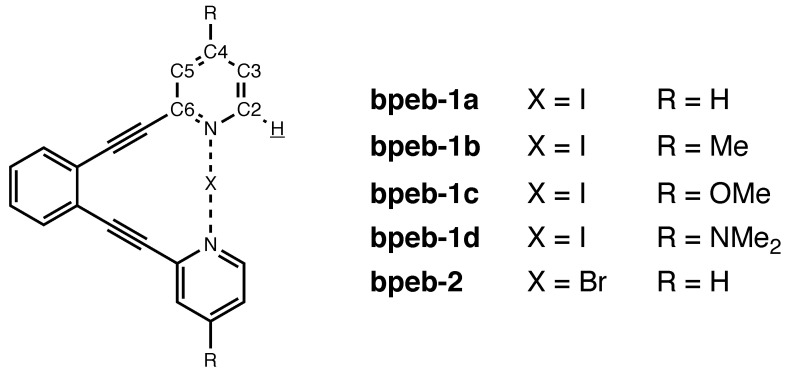
Structures and numbering of atoms for the molecules considered in Section 3.2. The substitution sites are underlined.

**Figure 5 molecules-25-02915-f005:**
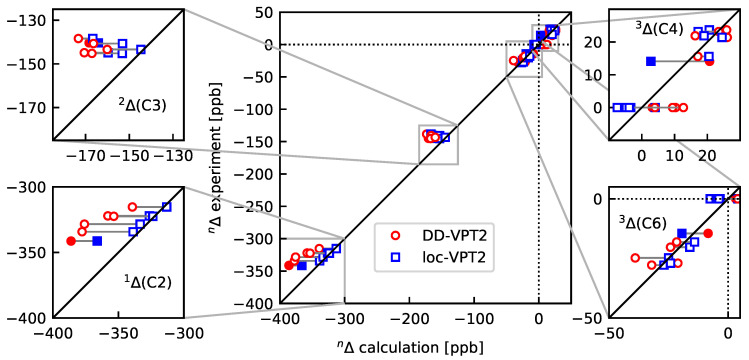
${}^{13}C$ NMR isotopic shifts upon a H/D substitution at H2 in the molecules considered in Section 3.2, calculated at the DD-VPT2 and loc-VPT2 levels of theory. The full markers indicate the values for pyridine, the hollow markers, those for **bpeb-1a** to **bpeb-1d.** See Section 4 for computational details.

**Figure 6 molecules-25-02915-f006:**
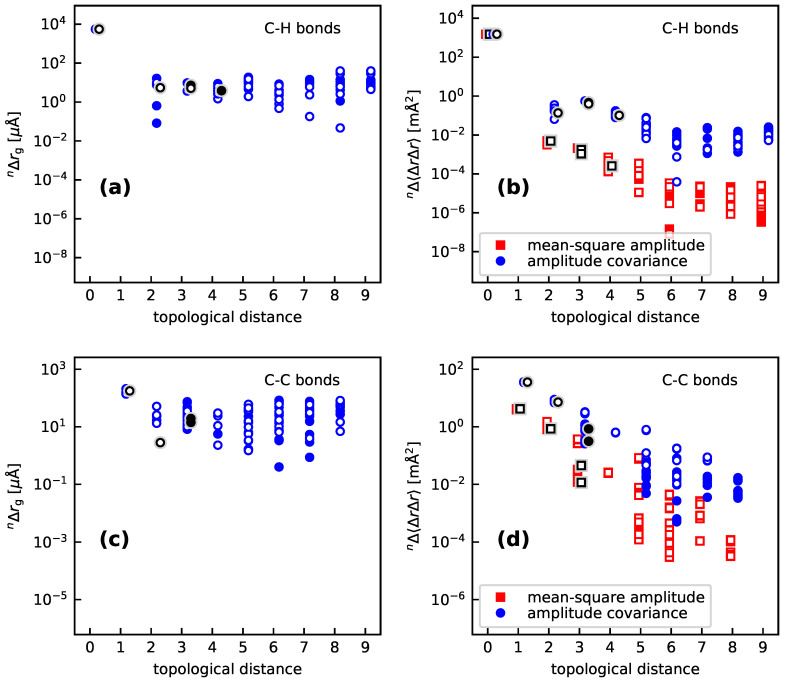
Isotope effects on the C–H (**a**) and C–C (**c**) bond lengths, as well as the mean-square vibration amplitudes and the covariance of the amplitudes with ${}^{n}\Delta r\left(C\underline{H}\right)$ for C–H (**b**) and C–C (**d**) bonds, for all H/D isotopic substitutions in the molecules investigated in Section 3.2. The topological distance is the number of bonds between the present bond and the H site, that is, 0 for the C–H bond itself, 1 for geminal bonds, 2 for vicinal bonds and so forth. The graphs show the absolute values of the isotope effects, full and hollow markers indicate positive and negative values, respectively. The black markers with grey shadings show the values for pyridine. See Section 4 for computational details.

**Figure 7 molecules-25-02915-f007:**
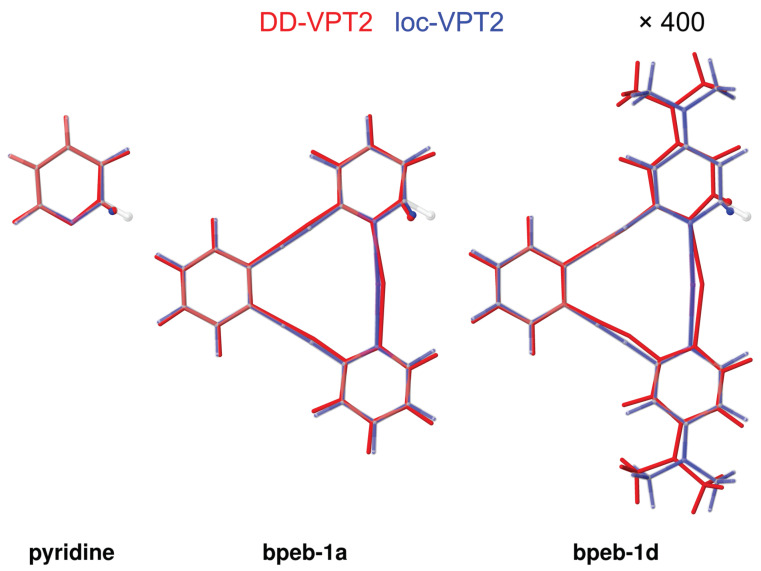
Isotope effects on the $r_{z}$ geometries of pyridine, **bpeb**-1a, and **bpeb-1d** by a single H/D substitution. The semitransparent structures show the $r_{e}$ geometries, red and blue structures show the geometry changes as calculated with DD-VPT2 and loc-VPT2, respectively. The substitution sites are marked by atomic spheres. Geometry changes enhanced by a factor of 400. Section 4 for computational details.

**Figure 8 molecules-25-02915-f008:**
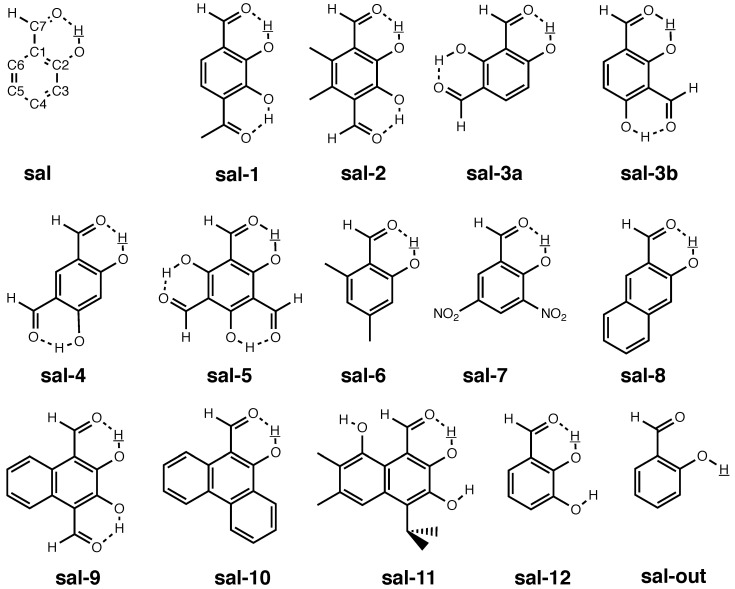
Structures and numbering of atoms for the molecules considered in Section 3.2. The substitution sites are underlined. **sal-out** is a hypothetic conformer of **sal** used for reference purposes.

**Figure 9 molecules-25-02915-f009:**
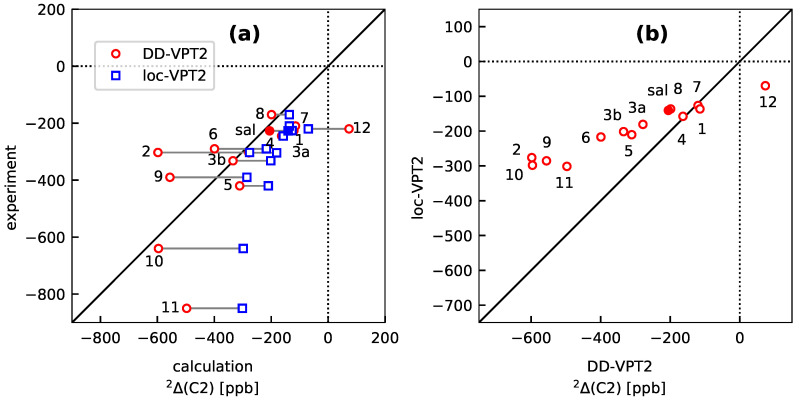
NMR isotope shifts ${}^{2}\Delta \left(C2\right)$ for **sal** and **sal-1** to **sal-12** as calculated with DD-VPT2 and loc-VPT2, respectively. (**a**) DD-VPT2 and loc-VPT2 versus experimental values, (**b**) loc-VPT2 vs. DD-VPT2 values. The **sal** derivatives are denoted with their numbers, for example, **3a** stands for **sal-3a**. The full markers are for the stem compound **sal**. See Section 4 for computational details.

**Figure 10 molecules-25-02915-f010:**
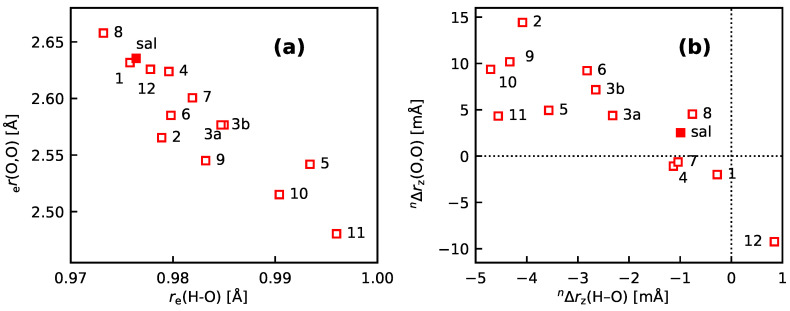
Calculated geometry parameters for **sal** and **sal-1** to **sal-12**. (**a**) $r_{e}\left(O,O\right)$ vs. $r_{e}\left(O–\underline{H}\right)$, (**b**) ${}^{n}\Delta r_{z}\left(O,O\right)$ vs. ${}^{n}\Delta r_{z}\left(O–\underline{H}\right)$ as calculated with DD-VPT2. The **sal** derivatives are denoted with their numbers, for example, **3a** stands for **sal-3a**. The full markers are for the stem compound **sal**. See Section 4 for computational details.

**Figure 11 molecules-25-02915-f011:**
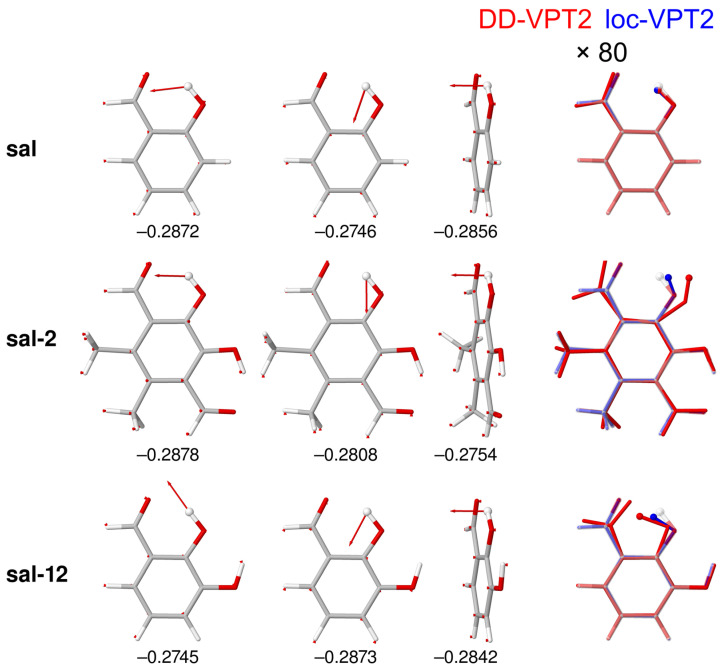
The three leading difference-dedicated vibration modes (first three columns) and isotope effects on the $r_{z}$ geometries (rightmost column) of **sal**, **sal-2**, and **sal-12** for a single H/D substitution in the intra-molecular hydrogen bond. The substitution sites are marked by atomic spheres. For the difference-dedicated vibration modes, the numbers below the structures give the corresponding weight factors $\kappa_{i}$. In the rightmost column, semitransparent structures show the $r_{e}$ geometries, red and blue structures show the geometry changes as calculated with DD-VPT2 and loc-VPT2, respectively. Geometry changes enhanced by a factor of 80. See Section 4 for computational details.

**Figure 12 molecules-25-02915-f012:**
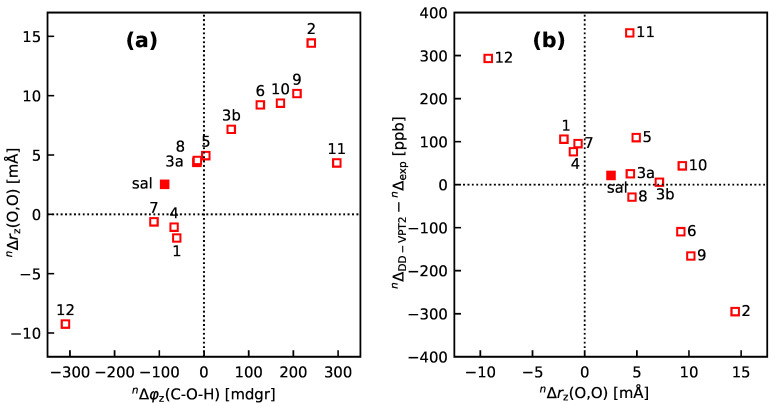
(**a**) ${}^{n}\Delta r_{z}\left(O,O\right)$ vs. **${}^{n}\Delta \varphi_{z}\left(C–O–H\right)$** and (**b**) the difference ${}^{n}\Delta {\left(C2\right)}_{\mathrm{DD}-\mathrm{VPT}2}-{}^{n}\Delta {\left(C2\right)}_{\exp}$ vs. **${}^{n}\Delta r_{z}\left(O,O\right)$** for **sal** and **sal-1** to **sal-12**. The **sal** derivatives are denoted with their numbers, for example, **3a** stands for **sal-3a**. The full markers are for the stem compound **sal**. See Section 4 for computational details.

**Table 1 molecules-25-02915-t001:** Isotope shifts for a H/D substitution at H2 in pyridine calculated at various levels of theory and mean signed (MSgD) and root-of-mean-square (RMS) differences from experiment. Calculations done with ωB97X-D/pc-2 force field. DD-VPT2 values from Reference [30]. All values calculated for dichloromethane solution and given in ppb.

Method/							25 °C		0 K	
Condition		C2	C3	C4	C5	C6	MSgD	RMS	MSgD	RMS
DD-VPT2	ωB97X-D	−381.5	−168.1	5.8	24.4	−4.5	−11.9	25.8	1.4	13.0
	B3LYP	−416.1	−172.2	4.2	25.0	−3.8	−21.3	41.6	−7.9	27.1
loc-VPT2	ωB97X-D	−362.2	−165.1	4.9	7.6	−15.6	−13.3	16.7	0.0	10.1
	B3LYP	−399.0	−169.4	3.3	8.0	−15.2	−23.4	32.6	−10.1	18.8
LMZL	ωB97X-D	−310.4	-−137.1	8.1	7.2	−20.1	5.4	15.9	18.7	33.1
	B3LYP	−339.1	−138.6	6.6	7.9	−20.4	−2.0	4.2	11.3	21.5
LMZL+cent	ωB97X-D	−351.1	−138.8	6.8	7.7	−19.5	−4.9	6.4	8.4	18.3
	B3LYP	−387.6	−140.8	5.3	8.3	−19.6	−14.4	23.6	−1.1	18.5
experiment										
25 °C	[36]	−341.0	−140.0		14.0	−15.0				
0 K (extrapolated)	[30]	−367.1	−166.4		24.2	−26.0				

**Table 2 molecules-25-02915-t002:** Mean signed and root-of-mean-square deviations and regression coefficients for the isotope shifts in norbornane and adamantane (see Section 3.1) and **bpeb**-derivative complexes (see Section 3.2). DD-VPT2 values from Ref. [30]. See Section 4 for computational details. Deviations given in ppb.

			MSgD	RMS	$r^{2}$
**Norbornane**		DD-VPT2	−16.8	26.9	0.9993
		loc-VPT2	−9.9	20.4	0.9960
**Adamantane**		DD-VPT2	−19.9	34.5	0.9994
		loc-VPT2	−18.3	37.2	0.9984
**bpeb derivatives**	25 °C	DD-VPT2	−13.9	23.8	0.9977
		loc-VPT2	12.4	20.6	0.9960
	0 K	DD-VPT2	2.0	12.9	0.9932
		loc-VPT2	−6.2	10.7	0.9916

**Table 3 molecules-25-02915-t003:** Isotope effects on selected bond distances and amplitudes in pyridine, benzene, and cyclohexane as calculated with different methods. Calculations done at the ωB97X-D/pc-2 level of theory. All values given in μÅ.

		Geometry		Amplitude (Mean Square)					
		$r(C\underline{H})$		$r(C\underline{H})$		$\varphi_{\mathrm{az}}$		$\varphi_{\mathrm{alt}}$	
		$r_{z}$	$r_{g}$	All-H	IE	All-H	IE	All-H	IE
**Pyridine**	DD-VPT2	−989	−5437	5703	−1525	39.0	−9.8	84.0	−12.9
	loc-VPT2	−1367	−5719	5468	−1600	35.5	−10.4	54.6	−16.0
	LMZL		−4427	5467	−1593	35.7	−10.4	54.6	−15.9
	LMZL+cent		−5407	5440	−1580	35.7	−10.4	54.6	−15.9
**Benzene (H/D)**	DD-VPT2	−769	−5369	5655	−1512	41.1	−10.3	85.2	−13.6
	loc-VPT2	−1132	−5652	5425	−1588	37.7	−11.0	56.1	−16.4
	LMZL		−4319	5410	−1577	38.1	−11.1	57.4	−16.7
	LMZL+cent		−5252	5385	−1564	38.1	−11.1	57.4	−16.7
**Cyclohexane** **(axial)**	DD-VPT2	−1740	−5987	5957	−1584	41.3	−10.5	56.8	−11.5
	loc-VPT2	−2542	−6320	5695	−1667	37.7	−11.0	39.7	−11.6
	LMZL		−4997	5691	−1659	38.2	−11.1	39.7	−11.6
	Ref. [60]		−4700		−1900		−11.0		−11.6
	LMZL+cent		−6111	5653	−1641	38.2	−11.1	39.7	−11.6
**Cyclohexane** **(equatorial)**	DD-VPT2	−1563	−5851	5885	−1567	42.0	−10.7	58.0	−11.5
	loc-VPT2	−2341	−6169	5632	−1648	38.5	−11.3	40.2	−11.8
	LMZL		−4768	5601	−1633	38.9	−11.3	40.6	−11.8
	Ref. [60]		−4700		−1900		−11.2		−11.6
	LMZL+cent		−5854	5589	−1623	38.9	−11.3	40.6	−11.8

**Table 4 molecules-25-02915-t004:** Isotope shifts for a single ${}^{12}C{/}^{13}C$ substitution in benzene calculated at various levels of theory. Calculations done with the ωB97X-D XC functional and a pc-2 basis set for the force fields and a pcS-2 basis set for the NMR calculations, for acetone solution and given in ppb.

Method		H1	H2	H3	H4	MSgD	RMS
DD-VPT2	ωB97X-D	−2.46	−0.97	0.07	0.20	−0.14	0.23
	B3LYP	−2.39	−0.99	0.04	0.21	−0.13	0.20
loc-VPT2	ωB97X-D	−1.79	−0.76	0.17	0.05	0.07	0.17
	B3LYP	−1.74	−0.81	0.15	0.07	0.07	0.18
experiment [64]		−2.04	−0.81	0.02	0.22		

**Table 5 molecules-25-02915-t005:** Isotope effects on bond distances and mean-square vibration amplitudes for a single ${}^{12}C{/}^{13}C$ substitution in benzene. Calculations done with ωB97X-D/pc-2 force fields. Values for geometries given in μÅ, values for amplitudes in mÅ${}^{2}$ or dgr${}^{2}$, respectively.

	Geometry				Amplitude (Mean Square)							
	$r(\underline{C}H)$		$r(\underline{C}C)$		$r(\underline{C}H)$		$r(\underline{C}C)$		$\varphi_{\mathrm{az}}$		$\varphi_{\mathrm{alt}}$	
	$r_{z}$	$r_{g}$	$r_{z}$	$r_{g}$	All-H	IE	All-H	IE	All-H	IE	All-H	IE
DD-VPT2	−42	−67	−63	−119	5655	−16.4	2010	−38.5	41.1	−0.18	85.2	−1.18
loc-VPT2	17	−50	−8	−60	1088	−42.8	1137	−44.7	6.3	−0.25	46.0	−1.81

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
