# Peer review of "The Structure of the “Vibration Hole” around an Isotopic Substitution—Implications for the Calculation of Nuclear Magnetic Resonance (NMR) Isotopic Shifts"

_molecules, 2020, doi:10.3390/molecules25122915_

Round 1

Reviewer 1 Report

Gräfenstein uses his recently described difference-dedicated second-order vibrational perturbation theory method to investigate what he calls the ‘vibrational hole’ around hydrogen and carbon isotopic substitutions in a series of reference molecules. The paper is comprehensive and while it is quite long, it is fairly easy to follow. The way the DD-VPT2 method leads to a natural truncation of the relevant modes gives a useful speedup over the standard VPT2, which should allow the method to be applied to larger systems.

For anyone used to looking at animations of normal mode vibrations, Figures 3 and 11 are quite intuitive and the demonstration of the isotope effects on geometry fits with expectation. I wonder if these figures might benefit from an additional panel showing the relevant normal mode vectors mapped onto the equilibrium geometry?

I may have missed this, but has the author considered how the method would cope with multiple isotopic substitutions?

Author Response

I thank the referee for their constructive comments, which helped me to improve the ms.

Gräfenstein uses his recently described difference-dedicated second-order vibrational perturbation theory method to investigate what he calls the ‘vibrational hole’ around hydrogen and carbon isotopic substitutions in a series of reference molecules. The paper is comprehensive and while it is quite long, it is fairly easy to follow. The way the DD-VPT2 method leads to a natural truncation of the relevant modes gives a useful speedup over the standard VPT2, which should allow the method to be applied to larger systems.

For anyone used to looking at animations of normal mode vibrations, Figures 3 and 11 are quite intuitive and the demonstration of the isotope effects on geometry fits with expectation. I wonder if these figures might benefit from an additional panel showing the relevant normal mode vectors mapped onto the equilibrium geometry?

The isotopic effect on the vibration motif often involves a large number of normal modes for each isotopologue, particularly if the isotopic substitution affects the molecular symmetry, as is the case for benzene (D6h to C2v). [I discuss this point in more detail for benzene in the introduction of my paper in J. Chem. Phys. 151, 244120 (2019); https://doi.org/10.1063/1.5134538] (paper I in the following). A more suitable way to describe the isotopic effect on the vibration amplitudes are the difference-dedicated vibration modes (see Sec. 2.1, in particular Eq. 10 in the ms). I have therefore completed Figs. 3 and 11 with the most important difference-dedicated vibration modes for the respective compounds and isotopic substitutions. This change has been complemented by some additional changes in the text:

  1. The captions of Fig. 3 and Fig. 11 were changed accordingly.
  2. Lines 305 to 319: The paragraph in Sec. 3.1 that refers to Fig. 3 was rewritten. The difference-dedicated modes are discussed in brief. Also, it is mentioned that Molden files containing the difference-dedicated modes for all compounds discussed in the ms are found in the SI to paper I or the present ms.
  3. Lines 450 to 464: The paragraph in Sec. 3.3 that refers to Fig. 11 was rewritten to incorporate a discussion of the difference-dedicated modes.
  4. Molden files with the differnce-dedicated modes for the sal derivatives were added to the zip archive in the SI.
  5. The molden files are mentioned in Section S1 of the SI.

I may have missed this, but has the author considered how the method would cope with multiple isotopic substitutions?

The DD-VPT2 algorithm is applicable to any isotopic substitution. However, for multiple substitutions the number of relevant difference-dedicated vibration modes increases roughly proportionally to the number of substitution sites. This means that the relative gain compared to standard VPT2 decreases (but vill typically remain substantial).

Lines 130–134: I have added a brief discussion of this point in Sec. 2.

Reviewer 2 Report

The work entitled "The structure of the 'vibration hole' around an isotopic substitution. Implications for the calculation of NMR isotopic shifts " by Gräfenstein is a good manuscript that I suggest to publish in the present form.

More in detail, in this manuscript the author uses a previously developed theoretical protocol (difference-dedicated second-order vibrational perturbation theory method) to obtain accurate calculation of NMR isotopic shifts for a series of target organic systems.

The manuscript give new insights on the spectroscopic behavior of these systems. The employed methodology is clearly described, the results are well reported and discussed, and the conclusions are fully consistent with the obtained data.

The subject can be of interest for a wide range of readerships. I suggest the publication of this good work in the present form.

Author Response

The work entitled "The structure of the 'vibration hole' around an isotopic substitution. Implications for the calculation of NMR isotopic shifts " by Gräfenstein is a good manuscript that I suggest to publish in the present form.

More in detail, in this manuscript the author uses a previously developed theoretical protocol (difference-dedicated second-order vibrational perturbation theory method) to obtain accurate calculation of NMR isotopic shifts for a series of target organic systems.

The manuscript give new insights on the spectroscopic behavior of these systems. The employed methodology is clearly described, the results are well reported and discussed, and the conclusions are fully consistent with the obtained data.

The subject can be of interest for a wide range of readerships. I suggest the publication of this good work in the present form.

I thank the referee for their encouraging comments and hope that the audience of Molecules will benefit from the ms.